The relationship between ethylene-induced autophagy and reactive oxygen species in Arabidopsis root cells during the early stages of waterlogging stress

Zheng Qiwei
Li Gege
Wang Hongyan
Zhou Zhuqing zhouzhuqing@mail.hzau.edu.cn
Laboratory of Cell Biology, College of Life Science and Technology, Huazhong Agricultural University , Wuhan , Hubei , China
Hussain Saddam
Electronic publication date: 2023 May 26
Publication date: 2023
Volume: 11
Electronic Location ID: e15404
Received 2022 Dec 2; Accepted 2023 Apr 20
Copyright: ©2023 Zheng et al.
Copyright year: 2023
Copyright holder: Zheng et al.
License: This is an open access article distributed under the terms of the Creative Commons Attribution License, which permits unrestricted use, distribution, reproduction and adaptation in any medium and for any purpose provided that it is properly attributed. For attribution, the original author(s), title, publication source (PeerJ) and either DOI or URL of the article must be cited.
License URL: https://creativecommons.org/licenses/by/4.0/

Keywords: Ethylene, Autophagy, Reactive oxygen species, Waterlogging stress, Arabidopsis

Funding: National Natural Science Foundation of China 31871530 This work was supported by the National Natural Science Foundation of China (No. 31871530). The funders had no role in study design, data collection and analysis, decision to publish, or preparation of the manuscript.

==============================
The response of plants to waterlogging stress is a complex process, with ethylene playing a crucial role as a signaling molecule. However, it remains unclear how ethylene is initially triggered in response to waterlogging stress when plants are continuously waterlogged for less than 12 hours. Here, we have shown that ethylene-induced autophagy leads to the degradation of damaged mitochondria (the main organelles producing reactive oxygen species (ROS)) to reduce ROS production during oxidative stress in Arabidopsis thaliana, which improves the survival rate of root cells in the early stages of waterlogging stress. Waterlogging stress activated ethylene-related genes, including ACO2, ACS2, ERF72, ERF73, and EIN3, and ethylene content of plants increased significantly within 24 h of continuous waterlogging. As stress duration increased, increased amounts of ROS accumulated in Arabidopsis thaliana roots, and the activity of antioxidant enzymes initially increased and then decreased. Concurrently, the level of ethylene-induced autophagy, which participates in antioxidant defense, is higher in wild-type plants than in the octuple acs mutant cs16651 (acs2-1/acs4-1/acs5-2/acs6-1/acs7-1/acs9-1/amiRacs8acs11). Exogenous application of 1-aminocyclopropanecarboxylic acid (ACC), resulted in a more pronounced manifestation of autophagy in the stele of Arabidopsis roots. Compared with the waterlogging treatment group or the ACC treatment group, the waterlogging + ACC treatment can induce autophagy to occur earlier and expand the autophagic range to the epidermis of Arabidopsis thaliana roots. Overall, our results provide insight into the important role of ethylene-induced autophagy in enhancing the antioxidative capacity of Arabidopsis thaliana during the early stages of waterlogging stress. Furthermore, we suggest ethylene as a potential candidate for mitigating the deleterious effects caused by waterlogging in Arabidopsis thaliana.

Introduction

During the growth process, plants face various abiotic stressors such as drought, salt, and flooding stress. These stressors can cause energy and carbohydrate crises in plants, limit plant growth and development, and lead to a decline in crop yield and quality (Zhu, 2016). Soil flooding creates a hypoxic environment (Phukan, Mishra & Shukla, 2016). Plants undergo two important changes during waterlogging: an increase in ROS and the gaseous plant hormone ethylene. Waterlogging-tolerant plants use changes in O2, ROS, and ethylene as signals to induce adaptive processes (Voesenek & Sasidharan, 2013).

The ethylene content in Arabidopsis root tips increases rapidly after submergence (Perata, 2020). Ethylene insensitive protein 3 (EIN3) is an important transcription factor that mediates the ethylene response (Xie et al., 2015; Hartman et al., 2019). Multiple studies have shown that ethylene is critical in hypoxia and metabolic adaptation during plant flooding and can adapt to hypoxia by enhancing the stability of ethylene-responsive factor family VII (ERF-VIIs) (Chen et al., 2015a; Yao et al., 2017; Hartman, Sasidharan & Voesenek, 2021). ERFs are involved in hormones, development, metabolic adaptation, and abiotic stress responses. Ethylene response factor 74 (ERF74) can activate the expression of respiratory burst oxidase homolog D (RbohD) by binding to the promoter of RbohD to induce ROS production (Yao et al., 2017). ROS homeostasis in plants modulates metabolic responses, contributes to morphological fitness, and increases the likelihood of plant survival under flooding stress (Yeung, Bailey-serres & Sasidharan, 2019). In maize, ethylene can upregulate the expression of ZmEREB180, which is closely related to waterlogging tolerance (Yu et al., 2019).

ROS, as metabolic byproducts and secondary messengers, play an important role in growth, development, and programmed cell death in response to the environment (Suzuki et al., 2011; Liu & He, 2016). In higher plants, ROS are produced in various subcellular structures during plant growth and development, including mitochondrial respiration, chloroplast photosynthesis, peroxisomes, and NADPH oxidase (respiratory burst oxidase homologues) located on the plasma membrane (Waszczak, Carmody & Kangasjärvi, 2018). During waterlogging, the formation of ROS is a common physiological process in plants (Mu et al., 2007; Iskandar & Mahmood, 2010; Parlanti et al., 2011). When plants are exposed to various environmental stressors, the content of ROS in plants increases, causing damage to lipids, proteins, and DNA (Gill & Tuteja, 2010; Wang et al., 2018). Plants have evolved corresponding antioxidant systems to remove excess ROS and protect them from oxidative damage. For example, superoxide dismutase (SOD) converts superoxide anion radicals (O2⋅−) into H2O2 and catalase (CAT) converts H2O2 into H2O and O2. Ascorbate peroxidase (APX) relieves the toxic side effects of H2O2, and glutathione reductase (GR) participates in metabolic regulation and antioxidant processes (Morel & Barouki, 1999; Foyer & Noctor, 2003; Iskandar & Mahmood, 2010).

Autophagy is a conserved process that decomposes misfolded proteins or toxic substances and damaged organelles into biological macromolecules (such as amino acid and fatty acids) for recycling to maintain eukaryotic growth, development, and survival (Li & Vierstra, 2012; Wang, Mugume & Bassham, 2018). Normally, autophagy occurs at very low levels, but the process is activated under various environmental cues (Marshall & Vierstra, 2018). Studies have shown that to resist waterlogging stress, autophagy can balance energy and ensure basic metabolism in plants (Chen et al., 2015b). Mitophagy is an adaptive regulation that is critical for reducing ROS levels and preventing cell death when mammalian cells are chronically hypoxic (Zhang et al., 2008). Although mitophagy is less frequently reported in plants, it can recycle damaged mitochondria during abiotic stress (Ma et al., 2021; Nguyen & Lazarou, 2021). Studies have shown that autophagy plays an important role in ethylene-mediated drought tolerance. Autophagy induced by ROS levels, dependent on mitochondrial alternating oxidase (AOX), is critical for ethylene-mediated drought tolerance in tomato. ERF5 binds to the promoters of autophagy-related genes (ATGs) ATG8d and ATG18 h and is critical for ethylene-induced autophagosome formation in tomato under drought conditions (Zhu et al., 2018). Another study showed that ethylene mediates the induction of GmATG8i in soybean plants under starvation stress (Okuda et al., 2011). Ethylene signaling is involved in autophagy in response to stress by inducing ATGs gene expression and altering ROS levels, but only in energy crises caused by insufficient carbohydrate reserves (Hartman, Sasidharan & Voesenek, 2021). Ethylene and ROS can enhance hypoxia resistance in wheat seedlings by controlling morphological adaptation and metabolic responses (Yamauchi et al., 2014). Under hypoxic conditions, root cortex cells induce RbohD translation to increase ROS production, which plays an important role in ethylene-induced stomatal formation in rice roots (Yamauchi et al., 2017). Some studies suggest that ethylene plays an important role in the induction of autophagy, thereby improving the survival rate under flooding and hypoxia stress (Hartman, Sasidharan & Voesenek, 2021).

While we speculate that ethylene may be involved in hypoxia-mediated autophagy under waterlogging stress, we also propose that in plants, mitophagy can be activated by ethylene-dependent ROS production, which may have a negative regulatory effect on ROS production.

Materials & Methods

Experimental materials and growth conditions

The wild-type (WT) Arabidopsis used in this study was Columbia-0 (Col-0). Rbohd/f, octet acs mutant (CS16651, acs2-1/acs4-1/acs5-2/acs6-1/acs7-1/acs9-1/amiRacs8acs11), ein2-5, p35S::EIN3-GFP, and ein3/eil1-1 (Lv et al., 2018) were presented by Professor Ding Zhaojun of Shandong University. ATG8epro::GUS (Guan et al., 2019) was obtained from our laboratory. Before germination, the seed surface was sterilized with 5% sodium hypochlorite NaClO + 0.05% TWEEN-20 solution for 12 min, washed with sterile water five times, and vernalized at 4 °C for 3 days. Then, the seeds were sown on 1/2 MS medium and cultured in a growth room (16/8 h light/dark at 22 °C /19 °C and 70% relative humidity) for 7 days.

Experimental design

Waterlogging Treatment (WL): According to the method (Wang et al., 2020), the waterlogging treatment was carried out with slight modification. Seven-day-old seedlings were transferred from 1/2 MS medium to a Petri dish filled with distilled water and placed in a growth chamber under controlled conditions so that the roots of Arabidopsis seedlings are completely immersed in distilled water. The leaves were kept 0.5 cm above the water surface.

ACC Treatment (ACC): Seven-day-old seedlings were transferred to 1/2 MS medium supplemented with 0.05 µM ACC (Sigma-Aldrich, St. Louis, MO, USA).

WLCACC Treatment (WLCACC): After diluting the ACC stock solution to a concentration of 0.05 µM with distilled water, 7-day-old seedlings were transferred from 1/2 MS medium to 1/2 MS medium containing 0.05 µM ACC. The seedlings were placed in a growth chamber with their roots completely submerged in the 0.05 µM ACC solution, while their leaves were kept 0.5 cm above the water surface. All treatments were performed in three independent biological replicates.

Ethylene measurements

The method (Sun et al., 2017) was slightly modified to determine the ethylene content, as described above. Seven-day-old seedlings were randomly selected, with 30 of them transferred to a 25 mL vial containing three mL of liquid MS medium. The vial was then sealed with a rubber stopper and left for 12 h at 22 °C before measuring the ethylene content. A gas-tight syringe was used to draw one mL of the upper air sample from each bottle through the septum, which was immediately injected into a GDX-502 column (2 m * 1/8 inch) and flame ionization detector (FID) gas chromatograph (Agilent 7890BGC, Agilent Technologies, Santa Clara, CA, USA). Nitrogen (N2) was used as the carrier gas and separation was performed at 90 °C. The ethylene peak area was integrated with Agilent ChemStation, and the result was expressed as the average relative ethylene production (%) of each treatment.

ROS assays

Nitro Blue Tetrazolium Chloride monohydrate (NBT) and 3,3′-diaminobenzidine (DAB) staining were performed according to the method (Kumar et al., 2014) with minor modifications. Seven-day-old seedlings were immersed in 2 mg/mL NBT solution (Sigma-Aldrich) for 15 min and then washed three times with distilled water. The seedlings were photographed with a differential interference microscopy (Nikon 80i Eclipse) and the staining intensity was analyzed using Image J software to detect O2⋅− content. To detect H2O2 content, the seedlings were immersed in a 1 mg/mL DAB solution (Sigma-Aldrich) for 2 h, washed 3 times with distilled water, and then photographed with a differential interference microscopy (Nikon 80i Eclipse) and the staining intensity was analyzed using Image J software to detect H2O2 content. ROS level was detected by H2DCF-DA staining. Seven-day-old seedlings were immersed in 10 µM H2DCF-DA solution for 30 min and then washed 3 times with distilled water. The seedlings were photographed with a laser scanning confocal microscopy (TCS SP8, Leica, Wetzlar, Germany) using 488 nm excitation light and 525 nm emission light, and the fluorescence intensity was analyzed using Image J software.

Observing autophagosomes in roots

To detect autophagy levels, seven-day-old seedlings were immersed in a 0.05 mM Monodansylcadaverine (MDC) solution (30432; Sigma-Aldrich), gently shaken, and protected from light for 10 min. The seedlings were then washed twice with phosphate-buffered saline (PBS) for 5 min each. Finally, the seedlings were photographed using a laser scanning confocal microscope (TCS SP8, Leica) with an excitation wavelength of 345 nm and an emission wavelength of 455 nm.

For the GUS staining assay, seven-day-old ATG8epro::GUS seedlings were soaked in 10 mL of GUS staining solution (0.1 M PBS (pH 7.0), 5 mM potassium ferricyanide, 5 mM potassium ferrocyanide, 0.1% Triton X-100, 0.5 mg/mL X-Gluc) for 10 min in a vacuum system. The samples were then incubated in 75% ethanol for 1 h at 37 °C in the dark. Finally, the seedlings were photographed using differential interference microscopy (Nikon 80i Eclipse).

Antioxidant enzyme activity measurement

CAT enzyme activity was calculated by measuring the absorbance at 405 nm, and SOD enzyme activity was calculated by measuring the absorbance at 550 nm. The roots of seven-day-old seedlings were collected and weighed, and the enzymatic activity of CAT and SOD was examined using a kit (Jiancheng Institute of Bioengineering, Nanjing, China) according to the manufacturer’s instructions.

RNA extraction and qRT-PCR

TRIzol reagent (15596026; Invitrogen, Waltham, MA, USA) was used to extract total RNA from the fresh roots of seven-day-old Arabidopsis seedlings. The cDNA template was synthesized using the PrimeScript RT kit (Takara, RR047A), and qRT-PCR was performed using the Hieff® qPCR SYBR Green Master Mix (Shanghai Yisheng Biotechnology Co., Ltd., Shanghai, China) according to the manufacturer’s instructions. The qRT-PCR primer sequences are provided in Appendix S1. The AtACTIN2 gene was used as an internal control, and relative gene expression was calculated using the 2−ΔΔCT method.

Transmission electron microscopy

Prepare TEM samples with minor modifications according to the method (Guan et al., 2019). Arabidopsis root meristems were selected and immediately cut into 3 mm long small segments, then fixed with 2.5% glutaraldehyde in 0.1 M PBS (pH 7.0) at 4 °C for 12 h. The samples were then rinsed three times with 0.1 M PBS for 10 min each time. After being fixed with 1% osmium tetroxide for 3 h, the samples were again rinsed three times with 0.1 M PBS for 10 min each time. The samples were dehydrated in a graded acetone series of 30%, 50%, 70%, 80%, 90%, and 100% (three times each, for 15 min per step) and embedded in SPI-PON 812 resin. Ultrathin sections (70 nm) were prepared on an ultramicrotome (Leica EM UC7, Germany) with a diamond knife and collected on Formvar-coated grids. To enhance the image contrast, the sections were stained with 2% uranyl acetate and lead citrate. Finally, the cell ultrastructure was observed using a transmission electron microscopy (Hitachi H-7650, Tokyo, Japan).

Cell viability test

Seven-day-old seedlings were soaked in 10 mL propidium iodide (PI) solution for 10 min and photographed with a laser scanning confocal microscopy (TCS SP8, Leica) using an excitation wavelength of 488 nm and an emission wavelength of 660 nm. Seven-day-old seedlings were soaked in 20 µM fluorescein diacetate (FDA) solution and treated at 37 °C for 10 min. Subsequently, the seedling were photographed with a laser scanning confocal microscopy (TCS SP8, Leica) using an excitation wavelength of 480 nm and an emission wavelength of 530 nm.

Statistical analysis

All the experiments were performed in triplicate or more unless otherwise indicated, and the results reported in this study are presented as the mean ± SD. Data were analyzed by a two-tailed Student’s t-test using GraphPad Prism 7.0. The significance levels are * P < 0.05, ** P < 0.01, and *** P < 0.001.

Results

Ethylene responds to waterlogging stress

To investigate the physiological regulation of ethylene in plants under waterlogging conditions, an experiment was conducted to screen the ACC concentration, which revealed that 0.5 µM ACC significantly inhibited the growth of Arabidopsis roots (Fig. 1A). Seven-day-old WT seedlings were then subjected to ACC, WL, and WL+ACC treatments. The root lengths of the treated groups were all significantly shorter compared to the control group (Fig. 1B), indicating that both WL and ACC treatments negatively affected root growth. To analyze the changes in ethylene content in plants after waterlogging, the relative ethylene content was measured at different time points in the WT (Fig. 1C). The ethylene content in the WL treatment group increased significantly at 24 h compared to the control group. The WL+ACC treatment group had the highest ethylene content at 8 h compared to the control group. The ethylene content in the ACC treatment group varied significantly at 4 h and reached its maximum at 12 h. These results demonstrate that the ethylene content in Arabidopsis can be increased under waterlogging stress. To investigate the molecular mechanism of plant ethylene response to waterlogging stress, the expression levels of genes ACO2, ACS2, ERF72, ERF73, and EIN3, which are closely related to ethylene signaling, were analyzed. The relative expression levels of these genes were significantly increased in the WL and WL+ACC treatment groups, indicating that waterlogging stress can enhance the expression of ethylene signaling (Fig. 1D). ERF72 and ERF73, which are ethylene-responsive factors associated with hypoxic stress, were significantly upregulated in the WL and WL+ACC treatment groups. ACO2 and ACS2, which are involved in the conversion of ACC to ethylene, were significantly upregulated in the early stages of the ACC treatment group. EIN3 is an important gene downstream of the ethylene signaling pathway that is involved in ethylene-induced biological responses. The p35S::EIN3-GFP plants were observed using laser confocal microscopy under different experimental treatments. Compared to the control group, the fluorescence signals of the WL and ACC treatment groups were significantly enhanced. Interestingly, the fluorescence intensity of the WL+ACC group decreased at 12 h compared to the WL and ACC-treated groups (Fig. S1). Therefore, both WL and ACC treatments can induce EIN3 protein synthesis and regulate related physiological changes.

Figure 1 Phenotypic analysis of 7-day WT seedlings under different treatments (A–D).

(A) The phenotype of 7-day-old WT seedlings under different concentrations of ACC solution, scale bar = 1 cm. (A) Statistical analysis of the root length in Fig. 1A. (B) Phenotypic changes in root length of 7-day-old WT seedlings under different treatments for 48 h, scale bar = 1 cm. (B) Statistical analysis of the root length in Fig. 1B. (C) Quantification of the relative ethylene production (compared to 0 h) of 7-day-old WT seedlings under different treatments at different time points. (D) QRT-PCR analyses showing the relative expression of ethylene-related genes (ACO2, ACS2, ERF72, ERF73 and EIN3) in 7-day-old WT seedlings under different treatments at different time points. All of the experiments were performed for three biological replications. Data shown are the mean ± SD (n = 3). * P < 0.05; ** P < 0.01; *** P < 0.001 by Student’s t-test.

Ethylene induces partial ROS production and activates antioxidant enzyme system

Previous studies have shown that waterlogging stress can increase ROS accumulation in Arabidopsis roots (Guan et al., 2019). To determine whether there is a regulatory relationship between ethylene and ROS under waterlogging stress, we used H2 DCFH-DA, DAB, and NBT staining methods to detect the contents of ROS, H2O2, and O2⋅− in WT roots after different treatments (Fig. 2A). Compared to 0 h, the ROS content in the WL treatment group was significantly increased at 24 h, and the ROS content in the ACC treatment group reached its maximum at 12 h (Figs. 2A–2D). The contents of ROS, H2O2, and O2⋅−in ACC and WL+ACC treatment groups were significantly increased. Compared to the WL and ACC treatment groups, the WL+ACC treatment group showed an earlier and more significant increase in ROS and H2O2 content (Figs. 2A–2D, 2A–2F). To further demonstrate the role of ethylene in inducing ROS, cs16651 mutants were submerged, and the ethylene content was measured. The ethylene content produced by cs16651 mutants under waterlogging stress for 12 h showed no significant difference compared to 0 h (Fig. S2B). With an increase in waterlogging time, the H2O2 content of the cs16651 mutants continuously increased (Fig. S2A). The H2O2 content was significantly lower than that of the WT in the same treatment group (Fig. S2A). The H2O2 content in ein3/eil1-1 plants was continuously increased (Fig. S2A). The H2O2 content of ein3/eil1-1 was significantly lower than that of the WL treatment group at the same time point in WT (Fig. S2A). RBOHD and RBOHF enzymes play important roles in ROS production. Under the waterlogging treatment, the ethylene content of rbohd/f mutants was not significantly different from that of the WT at the same time point (Fig. S2B). In the 0–24 h waterlogging period, the H2O2 content of the rbohd/f mutants was lower than that of the WT in the same period but still produced a large amount of H2O2 (Fig. S2A). These experiments further demonstrated that ethylene production in Arabidopsis promotes ROS generation under waterlogging stress. When plants are subjected to environmental stress, the balance of ROS is disrupted, affecting the normal metabolism and regulatory functions of plants. As the ROS content of the treatment group increased, the relative expression levels of the CAT1 and SOD1 genes were up-regulated (Fig. 2B). During the treatment period from 0 to 24 h, the activities of CAT1 and SOD1 in Arabidopsis were higher in the WL+ACC treatment group before 12 h (Fig. S3), indicating an earlier antioxidant role and ROS scavenging. The expression of the GST1 gene was mainly up-regulated after 12 h of submersion, further exerting an antioxidant effect (Fig. 2B).

Figure 2 Dynamics of ROS, H2O2 and O2⋅− changes in Arabidopsis roots under different treatments (A–B).

(A) ROS fluorescence probe H2 DCFH-DA was used to detect the changes of ROS in the roots of 7-day-old WT seedlings under different treatment, scale bar = 25 µm. (B) DAB staining was used to detect the changes of H2O2 in the roots of 7-day-old WT seedlings under different treatment, scale bar = 50 µm. (C) NBT staining was used to detect the changes of O2⋅− in the roots of 7-day-old WT seedlings under different treatment, scale bar = 50 µm. (D–F) Relative staining intensities calculated from Fig. 2A, 2B and 2C. (B) QRT-PCR detected the relative expression of antioxidase related genes (SOD1, CAT1 and GST1) in 7-day-old WT seedlings under different treatments. All of the experiments were performed for three biological replications. Data shown are the mean ± SD (n = 3). * P < 0.05; ** P < 0.01; *** P < 0.001 by Student’s t-test.

Ethylene induces partial autophagy in stele under waterlogging stress

Autophagy is an important physiological process for plant energy reuse. ROS induce the production of autophagy when plants are waterlogged (Guan et al., 2019). ATG8epro::GUS seedlings were stained to observe the autophagy phenomenon in Arabidopsis roots under different treatments. In this study, autophagy appeared only in the stele in the ACC treatment group, and the number of autophagosomes gradually increased as the treatment was prolonged. In the WL treatment group, autophagy first occurred in the stele and then in the cortex. Autophagy appeared in the WL+ACC treatment group at 4 h in the stele and cortex, and the number of autophagosomes increased as the treatment time was extended (Fig. 3B). This experiment showed that autophagy in the root stele of Arabidopsis was induced by ethylene under waterlogging stress and that it occurred earlier than in the cortex. The MDC staining experiment showed that, compared to the control group, the number of autophagosomes in the three treatment groups of WT seedlings significantly increased after 24 h (Fig. 3D). Autophagy gene expression was significantly up-regulated in the ACC, WL, and WL+ACC treatment groups. However, during the period from 0 to 48 h, the relative expression levels of ATG2, ATG5, ATG7, ATG8e, and ATG10 in the WL+ACC group appeared earlier than those in the WL group (Fig. 3A), further indicating that ethylene plays a role in inducing autophagy.

Figure 3 Ethylene induces autophagy in Arabidopsis roots under flooding stress (A–C).

(A) QRT-PCR detected the relative expression of autophagy-related genes (ATG2, ATG5, ATG7, ATG8e, and ATG10) in 7-day-old WT seedlings under different treatments. (B) The expression of the ATG8epro::GUS reporter was monitored in WT roots under different treatments, scale bar = 50 µm. (C) Autophagosomes in roots of 7-day-old WT seedlings were observed by confocal microscopy after MDC staining under different treatments, scale bar = 75 µm. (D) The data is according to the statistics of the number of autophagosomes in Fig. 3C. All of the experiments were performed for three biological replications. Data shown are the mean ± SD (n = 3). * P < 0.05; ** P < 0.01; *** P < 0.001 by Student’s t-test.

Ethylene-induced autophagy helps scavenge ROS produced by mitochondria

To further elucidate the relationship between ethylene and autophagy under waterlogging stress, autophagosomes were observed using MDC staining in ein3/eil1-1, cs16651, and rbohd/f mutants after waterlogging (Fig. 4A). At 0 h, the number of autophagosomes in ein3/eil1-1, cs16651 and rbohd/f mutants was significantly lower than in the WT. However, the number of autophagosomes in ein3/eil1-1 and cs16651 mutants increased after 24 h of waterlogging and had no significant difference compared to the WT (Figs. 4A–4a). To further explore the role of ethylene-induced autophagy in plant responses to waterlogging stress, WT seedlings were observed by transmission electron microscopy. In the control group, cell morphology was normal, and the cell wall, cell membrane, and mitochondrial morphology were intact. However, in the other treatment groups, with the increase in treatment time, most of the mitochondrial cristae were degraded, and some mitochondrial cristae had completely disappeared to form vacuolar structures, and the number of mitochondria gradually decreased (Fig. 4B). In the early stage of waterlogging (duration of continuous waterlogging less than 12 h), the WL+ACC treatment group produced more autophagosomes, and more damaged mitochondria were cleared earlier (Figs. 4A–4b). After 24 h, vacuolated mitochondria in the cytoplasm were transported into vacuoles via the autophagy pathway (Fig. 4B). In addition, ethylene-related mutants ACS2, ACS6, ein3/eil1-1, and cs16651 were waterlogged and then observed by electron microscopy (Fig. S4A), and the number of mitochondrial lesions in the mutants was higher when waterlogged for 4 h (Fig. S4a).

Figure 4 Observation on the ultrastructure of Arabidopsis root cells in different treatments (A–B).

(A) WT seedlings, ein3/eil1-1, CS16651, and rbohd/f mutants seedlings were stained with MDC under waterlogging treatment and then were used to observe autophagosomes, scale bar = 75 µm. (A) The data is according to the statistics of the number of autophagosomes in Fig. 4A. (B) That ultrastructure of the roots of WT seedlings under different treatments was observed with a transmission electron microscopy. Arrows indicate autophagosomes or autophagic structures. CW, cell wall; M, mitochondrion or degraded mitochondria; ER, Endoplasmic reticulum; G, Golgi; V, Vacuole. Scale bar = 1 µm. (B) The data is according to the statistics of the number of autophagosomes and the ratio of damaged mitochondria to intact mitochondria in a single cell in Fig. 4B. All of the experiments were performed for three biological replications. Data shown are the mean ± SD (n = 3). * P < 0.05; ** P < 0.01; *** P < 0.001 by Student’s t-test.

Effects of waterlogging on root cell activity

To further explore the physiological changes of plants in response to waterlogging stress, WT seedlings were subjected to different treatments for PI staining and FDA staining to observe changes in root cell activity. PI staining showed that the root cells did not die significantly at 0–24 h (Fig. S5A), while FDA staining showed that the WL+ACC treatment group had higher cell viability than the WL treatment group in the early stage of waterlogging (Fig. S5B). However, long-term waterlogging stress leads to cell death and production of aerenchyma (Fig. S6).

Discussion

WL and ACC enhance ethylene signaling and negatively regulate Arabidopsis root growth

Climate disasters, such as waterlogging, drought and high temperature, often cause huge losses to agriculture. To resist waterlogging, plants have evolved complex mechanisms to adapt to flooded environments. Studies have shown that ethylene not only rapidly accumulates in plant tissues but also rapidly activates ethylene-induced physiological responses under flooding stress conditions (Hartman, Sasidharan & Voesenek, 2021). Deep water induces the expression of genes SNORKEL1 and SNORKEL2, which encode ethylene response factors involved in ethylene signaling in rice and indirectly increase the length of rice internodes to enhance gas exchange with water to adapt to flooding stress (Hattori et al., 2009). The interrelationship between ethylene and ROS produced by plants under submerged conditions has been reported less frequently in Arabidopsis. Compared to the control group, the ACC and WL-treated group showed a significant increase in the number of damaged mitochondria in root cells (Fig. 4B), which may have contributed to the relatively stunted growth observed in the treated seedlings due to the crucial role of mitochondria in plant growth, suggesting negative regulation of root growth by WL and ACC (Fig. 1B). It is speculated that the difference in root length between the different treatment groups may be related to the ethylene content. To further explore the changes in ethylene content after WL treatment, this study showed that the ethylene content of the WL treatment group and ACC treatment group increased continuously with the treatment time. Compared with the WL and ACC treatment groups, the ethylene content of the WL+ACC treatment group did not increase significantly, but ethylene accumulated rapidly in the early stage of treatment, reaching the maximum ethylene content at 8 h, and decreased after 12 h (Fig. 1C). Treatment with ACC in wheat seedlings can increase ethylene accumulation in the roots of wheat seedlings and enhance the seedlings’ tolerance to hypoxic conditions (Yamauchi et al., 2014). It has also been reported that ethylene promotes root hair growth in Arabidopsis by coordinating the activities of EIN3/EIL1 and RHD6/RSL1 (Feng et al., 2017). Salt and drought stress can induce the expression of MnEIL3, which is a homologous gene of ein3 in the roots and shoots of mulberry, and Arabidopsis overexpressing MnEIL3 showed enhanced tolerance to salt and drought stress. Therefore, MnEIL3 may play an important role in abiotic stress resistance (Liu et al., 2019). Using different treatments in Arabidopsis EIN3-GFP plants, the expression of the EIN3 protein in Arabidopsis was significantly up-regulated in the WL and ACC treatment groups, but the EIN3 protein in the WL+ACC treatment group was not significantly up-regulated at 24 h (Fig. S1). The qRT-PCR data of ethylene-related genes further demonstrated that WL treatment and ACC treatment enhanced ethylene signaling (Fig. 1D). The above experimental results indicate that ethylene may only accumulate and play a role in the early stage of waterlogging stress, and that plants may undergo adaptive morphological changes in the later stage of waterlogging to respond to stress. However, further studies are needed to determine which genes in the ethylene signaling pathway are involved in the regulation mechanism of plant waterlogging stress.

The relationship between ethylene, ROS and autophagy under waterlogging stress

Ethylene plays an important role in inducing autophagy and reducing ROS in organisms, thereby improving plant survival during flooding, hypoxia and reoxygenation stress (Hartman, Sasidharan & Voesenek, 2021). This study showed that ACC could rapidly induce ROS production in Arabidopsis under waterlogging stress (Fig. 2A). Additionally, studies have shown that the expressions of GmATG8i, GmATG4, GmACC synthase, GmERF, and GmEin3 are up-regulated in soybean seedlings under starvation stress, and starvation-induced autophagy may be partially mediated by ethylene signaling (Okuda et al., 2011). In this study, the expression of autophagy genes in Arabidopsis was up-regulated under waterlogging stress, and the phenomenon of autophagy first appeared in the vascular column. Over time, the phenomenon of autophagy was also found in the cortex of the root. Autophagy occurred only in the vascular column in the roots under ACC treatment. Compared with the WL and ACC-treated groups, the location and extent of autophagy in the WL+ACC treatment group were changed (Fig. 3B). Various abiotic stressors, such as high salt, high temperature, drought, low temperature, and hypoxia can induce autophagy, which plays an important role in balancing plant growth and enhancing stress tolerance (Qi, Xia & Xiao, 2021). Additionally, this study found that the number of autophagosomes in ein3/eil1-1, cs16651, and rbohd/f mutants were significantly lower than that of the WT at 0 h (Fig. 4A). Therefore, we presumed that this explains the difference in the number of autophagosomes and ROS content between the mutants (Fig. S2A).

Ethylene participates in the induction of ROS generation under waterlogging stress

Adventitious root formation is a response to plant adaptation to flooding stress, and ROS signaling is involved in ethylene- and auxin-induced adventitious root formation in flooded cucumber plants (Qi et al., 2020). However, the regulatory relationship between ethylene and ROS under waterlogging stress remains unclear. Previous studies in the laboratory have demonstrated that roots can produce ROS after waterlogging, and ROS is an important signal in response to plant programmed cell death (Guan et al., 2019). WL and ACC treatments induced ROS production in Arabidopsis (Fig. 2A). The ethylene content no longer increased after 12 h (Fig. 1C), and the ROS content began to decrease in the ACC treatment group. The ROS content is closely related to the ethylene content, and ACC treatment increased the O2⋅− content in the roots (Figs. 2A–2c). Arabidopsis treated with AVG (2-aminoethoxyvinyl glycine, an inhibitor of ethylene synthesis) had a reduced O2⋅− content in the roots (Lv et al., 2018). After the WL+ACC treatment, the ethylene content and ethylene-related gene expression of WT plants decreased with the increase in treatment time(Figs. 1C and 1D), but the ROS content generally increased (Fig. 2A). The ethylene content of cs16651 mutant plants did not change significantly under waterlogging (Fig. S2B) but still produced ROS (Fig. S2A). Therefore, under waterlogging conditions, ROS production has a certain dependence on ethylene, but the increase in ROS is not completely dependent on ethylene. The decrease in ethylene content may also be due to excessive ROS levels, which negatively regulates ethylene production. However, the relevant regulatory mechanisms need further study.

Excessive accumulation of ROS activates the antioxidant system

Under normal circumstances, ROS in plants is maintained at a low level. When plants are subjected to external environmental stress, ROS content increases, affecting the normal metabolic microenvironment in plants (Wang et al., 2018). Plants have evolved a relatively complete scavenging system to maintain ROS balance (Gill & Tuteja, 2010). The activities of SOD, CAT, APX, and GR of the submerged-tolerant varieties were higher than those of the non-submerged varieties after 8 days of flooding treatment on rice. Rice can enhance the tolerance of rice to flooding stress by improving its antioxidant system (Iskandar & Mahmood, 2010). In the present study, compared with the WL treatment group, the WL+ACC-treated group had relatively higher CAT1 and SOD1 enzymatic activities at the early stage of waterlogging (Fig. S3), and CAT1 and SOD1 expression also occurred earlier (Fig. 2B), suggesting that excessive ROS accumulation activates the antioxidant system. During the 24–48 h period, the plants were in a waterlogged state, and the antioxidant system was insufficient to remove excess ROS. The WL and WL+ACC treatment groups continued to accumulate ROS. However, further studies are needed to determine which genes are involved in the regulation of the antioxidant system and lead to the weakening of the activity of antioxidant enzymes.

Conclusions

In summary, ethylene can promote ROS production in the early stages of waterlogging (duration of continuous waterlogging for less than 12 h), and plants remove excess ROS through the antioxidant system, resulting in a temporary decrease in ROS content. With prolonged waterlogging treatment time, the ROS content continued to rise (Fig. 2). When the antioxidant systems of CAT1 and SOD1 enzymes were insufficient to remove excess ROS in plants (Fig. S3), ethylene induced partial autophagy production, accelerating mitochondrial degradation to reduce ROS production and alleviate oxidative damage in plants (Fig. 4B). The autophagy pathway accelerates the degradation of mitochondria, which is essential for plants to alleviate oxidative damage and increase survival in the early stage of waterlogging stress. However, long-term waterlogging exacerbates mitochondrial degradation, resulting in cell death and the production of aerenchyma (Fig. S6). The lack of adequate energy supply in plants leads to restricted growth and development, and the shortened root length is a response to waterlogging stress (Fig. 1B). The mechanism by which ethylene induces autophagy and accelerates mitochondrial degradation is not fully understood. It is unclear to what extent ROS accumulation induces autophagy to degrade mitochondria. Additionally, why does ethylene-induced autophagy first occur in the vascular column and then spread to the root cortex after waterlogging? These issues warrant further study and exploration.

Supplemental Information

Figure S1 Expression of ethylene signaling in Arabidopsis roots under different treatments

Seven-day-old seedlings of p35S::EIN3-GFP were observed using confocal microscopy at 12 h under different treatments, scale bar = 75 µm. All of the experiments were performed for three biological replications.

Click here for additional data file.

Figure S2 Relationship between ROS and ethylene in mutants under waterlogging treatment (A–B)

(A) DAB staining was used to detect changes in H2O2 levels in the roots of WT, ein3/eil1-1, cs16651, and rbohd/f seedlings under waterlogging treatment, with a scale bar of 50 µm. (A) The relative intensity was calculated based on the results in Fig. S2A. (B) The relative ethylene production (compared to WT at 0 h) was quantified for 7-day-old WT, ein3/eil1-1, cs16651, and rbohd/f seedlings under waterlogging treatment. All of the experiments were performed for three biological replications. Data shown are the mean ± SD (n = 3). * P < 0.05; ** P < 0.01; *** P < 0.001 by Student’s t-test.

Click here for additional data file.

Figure S3 CAT and SOD activities in WT roots

The activities of CAT andSOD enzymeswere measured in 7-day-old WT seedlings under different treatments for varying durations. All of the experiments were performed for three biological replications. Data shown are the mean ± SD (n = 3). * P < 0.05; ** P < 0.01; *** P < 0.001 by Student’s t-test.

Click here for additional data file.

Figure S4 Observation on the ultrastructure of root cells of ethylene-related mutants Arabidopsis under waterlogging treatment

(A) The ultrastructure of roots from ethylene-related mutants ACS2, ACS6, ein3/eil1-1, and cs16651 seedlings was observed using transmission electron microscopy under waterlogging treatment at 0 h, 4 h, and 24 h. Arrows indicate autophagosomes or autophagic structures. CW denotes cell wall, M denotes mitochondrion or degraded mitochondria, ER denotes endoplasmic reticulum, G denotes Golgi, and V denotes vacuole. The scale bar is 1 µm. (A) The data are based on the ratio of damaged mitochondria to intact mitochondria in a single cell, as shown in Fig. S4A. (B) The data are based on the number of autophagosomes, as shown in Fig. S4A. All of the experiments were performed for three biological replications. Data shown are the mean ± SD (n = 3). * P < 0.05; ** P < 0.01; *** P < 0.001 by Student’s t-test.

Click here for additional data file.

Figure S5 Effects of different treatments on cell death, cell viability and morphology of roots.

(A) and (B) show cell death and cell viability in root cells of WT observed by PI and FDA staining, respectively, under different treatments, with a scale bar of 75 µm. All of the experiments were performed for three biological replications.

Click here for additional data file.

Figure S6 Effects of ethylene on plant roots under long-term waterlogging

Seven-day-old WT seedlings were treated for varying durations, and the morphological changes in WT root cells were examined through semi-thin section experiments. The sections were then photographed using differential interference microscopy (Nikon 80i Eclipse), with the aerenchyma formed by the cells indicated by the red star area. The scale bar is 50 µm. All of the experiments were performed for three biological replications.

Click here for additional data file.

Figure S7 Effects of ethylene on plant roots under different treatments

(A) Phenotypic changes in root length of 7-day-old WT seedlings under different treatments for 24 h, scale bar = 1 cm. (A) Statistical analysis of the root length in Fig. S7A.

Click here for additional data file.

Table S1 Primer used for qRT-PCR

Sequence of primers for several genes in RT-qPCR.

Click here for additional data file.

We thank Prof. Zhaojun Ding (Shandong University) for octet acs mutant (CS16651, acs2-1/acs4-1/acs5-2/acs6-1/acs7-1/acs9-1/amiRacs8acs11), ein2-5, p35S::EIN3-GFP, and ein3/eil1-1 seeds. We would like to thank LetPub for providing linguistic assistance during the preparation of this manuscript.

Additional Information and Declarations

Competing Interests

Author Contributions

Data Availability

The authors declare there are no competing interests.

Qiwei Zheng conceived and designed the experiments, performed the experiments, analyzed the data, prepared figures and/or tables, authored or reviewed drafts of the article, and approved the final draft.

Gege Li analyzed the data, authored or reviewed drafts of the article, and approved the final draft.

Hongyan Wang performed the experiments, prepared figures and/or tables, and approved the final draft.

Zhuqing Zhou conceived and designed the experiments, prepared figures and/or tables, and approved the final draft.

The following information was supplied regarding data availability:

The raw data is available at figshare: Zheng, Qiwei (2023): raw data. figshare. Dataset. https://doi.org/10.6084/m9.figshare.22318048.v1.

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
