# Peer review of "The relationship between ethylene-induced autophagy and reactive oxygen species in Arabidopsis root cells during the early stages of waterlogging stress"

_PeerJ, doi:10.7717/peerj.15404_

## Round 0.1 · original submission · Major Revisions

Although not all the comments from the referees were negative, the reviewers agree that your manuscript still requires additional work and major editing before being considered for publication. In the revised manuscript, clearly highlight the novel aspects of the study. You should improve the manuscript in order that it can be considered for a second round of revisions and further publication. Please provide an adequate response to each reviewer's comment and submit a list of responses to the comments indicating specifically in which lines the improvements were done, or explaining why they were not considered.

Reviewer 1 ·

Basic reporting

The manuscript by zheng et al., analyzed the “Relationship between ethylene-induced autophagy and ROS in Arabidopsis root cells under waterlogging stress”. While the result could be interesting, there are some weaknesses that the authors should address before publication.



Majors

1, Since the study focuses on the early stage of response, the author may modify the title as “XXX at the early stage”.



2, what is the ethylene concentration at the physiological level, for example with or without waterlogging stress? This could be important because the concentrations used in the study might be out of range in plants.



Minors

1, line120, what is 30 indicate in this sentence “Seven day old seedlings were randomly selected 30 and transferred to a 25 mL”?



2, line 191-208, why Figure1C and 1D appeared before 1B?

Experimental design

none

Validity of the findings

no

Additional comments

no

·

Basic reporting

The manuscript has studied the relationship between ET-induced autophagy and ROS in Arabidopsis root cells under waterlogging stress. Some interesting findings were reported and these findings were important for the related field. This reviewer has some comments as listed below.

The abstract can be improved in language writing. The writing of the section in conclusion is better than that of the result part in abstract.

Figure 1A was not cited in the text. The first part in the result (lines 194-196), the figure 1C and 1D should be named as figure 1A and 1B. The panels (A, B, C and D) in figure 1 should be rearranged accordingly.

As shown in figure 1A, the ET content was lower at 24h than the 12h in ACC treatment. Also, at 24h, the ET content in the WL+ACC treatment was lower than that in the WL treatment. Please give an explanation.

The supplemental figures have no figure legends in the manuscript.

Line 257, “treatment was the prolonged” should be “treatment was prolonged”.

Experimental design

Good.

Validity of the findings

no comment

Additional comments

no comment

---

## Round 0.2 · accepted · Accept

The authors have addressed all of the reviewers' comments. I have also assessed the revision by myself, and I am happy with the current version. This manuscript is ready now for publication.

Reviewer 1 ·

Basic reporting

The authors has adequately answered my questions. I am satisfied with the authors' response.

Experimental design

none

Validity of the findings

The authors has adequately answered my questions. I am satisfied withe the authors' response.

Additional comments

none

·

Basic reporting

I am satisfied with the revised manuscripts and has no more question

Experimental design

Good

Validity of the findings

Good

Additional comments

No more question